# Prevalence of antimicrobial resistance in Tanzania: A systematic review and meta-analysis

Charles Basil Kafaiya[1]*, Johnson Mshiu[1], Obadia Bishoge[1,2], Jonathan Mcharo[1], Sia Malekia[1], Irene Mremi[1], Angelina M. Lutambi[1], Mwanaada Kilima[3], Mary Mayige[1], Said Aboud[1,4]

1 National Institute for Medical Research (NIMR), Dar es Salaam, Tanzania, 2 Department of Environmental and Occupational Health, Muhimbili University of Health and Allied Sciences, Dar es Salaam, Tanzania, 3 Muhimbili National Hospital, Dar es Salaam, Tanzania, 4 Department of Microbiology and Immunology, Muhimbili University of Health and Allied Sciences, Dar es Salaam, Tanzania

* charles.kafaiya@nimr.or.tz

## Abstract

Antimicrobial resistance (AMR) threatens global health, and understanding resistance patterns aids in effective treatment and promotes responsible antimicrobial use. Despite the urgency of resistant pathogens, systematic reviews focusing specifically on Tanzania are limited, and while several studies report resistance patterns for individual pathogens, a consolidated analysis of overall prevalence is needed to inform policymaking and public health interventions. Therefore, this review and meta-analysis assessed the prevalence of antimicrobial resistance among clinically relevant pathogens in Tanzania, providing a comprehensive overview to support surveillance, infection control, and stewardship efforts. A total of 1865 studies identified from Google Scholar (1600), PubMed (13), and Science Direct (252) underwent screening and full article review. Finally, 28 studies were included. A subgroup analysis was performed to evaluate the resistance patterns within antibiotic classes for specific pathogens. Descriptive statistics were used to describe the characteristics of the studies, while the prevalence of antimicrobial resistance was estimated through Meta-analysis. Inconsistency and heterogeneity between studies were quantified by the $I^2$ index. Among the included studies, most isolates (25.0%) were obtained from urine samples. Of these studies, 75% were cross-sectional studies and 92.9% were conducted in hospital settings. The analysis revealed high resistance to penicillin, particularly amoxicillin-clavulanic and ampicillin, with *Klebsiella pneumoniae* (0.96 [0.83–0.99]), *Acinetobacter baumannii* (0.94 [0.67–0.99]) and Escherichia coli (0.90 [0.81–0.95]). Similarly, erythromycin resistance was most prevalent in *Campylobacter spp.* (0.85 [0.80–0.89]). Ciprofloxacin resistance was highest in *Acinetobacter baumannii* (0.54 [0.33–0.73]), whereas amikacin resistance was highest in *Proteus spp.* (0.86 [0.35–0.99]). Ceftriaxone resistance was particularly high in *Acinetobacter baumannii* (0.91 [0.70–0.98]) and *Pseudomonas aeruginosa* (0.85 [0.74–0.92]). Meropenem resistance was lowest among *Escherichia coli* (0.04 [0.01–0.10]) and

**Data availability statement:** All relevant data are within the paper and its Supporting Information files.

**Funding:** The author(s) received no specific funding for this work.

**Competing interests:** The authors have declared that no competing interests exist.

**Abbreviations:** ESKAPE-E, Enterococcus faecium, Staphylococcus aureus, Klebsiella pneumoniae, Acinetobacter baumannii, Pseudomonas aeruginosa and Enterobacter spp, Escherichia coli; STG, Standard Treatment Guideline; NEMLIT, The National Essential Medicines List of Tanzania; AMR, Antimicrobial resistance; BSI, bloodstream infections; UTI, Urinary Tract Infection; SSI, Surgical Site Infection; *AWaRe*, (*Access, Watch, Reserve*), *Access* (Penicillin, Erythromycin, Tetracycline, Nitrofurantoin) *Watch* (Cephalosporins, Ciprofloxacin, Sulphonamides), *Reserve* (Carbapenems, Cefepime, Clindamycin, Amikacin).

*Klebsiella spp.* (0.07 [0.03–0.15]), while the pooled resistance across ESKAPE-E pathogens was (0.11[0.06–0.19]). Imipenem and clindamycin each had an overall pooled resistance of (0.06[0.02–0.14]) against both *Escherichia coli* and *Klebsiella pneumoniae.* The findings highlight widespread resistance among bacterial pathogens, ESKAPE-E, particularly in the *Access* and *Watch* groups of antibiotics. The variability in resistance patterns underscores the need for the Ministry of Health to re-evaluate empirical treatment protocols (STG/NEMLIT) to ensure effective treatment regimens, strengthen antimicrobial stewardship, enhance surveillance systems, and promote rational antibiotic use.

## Introduction

Antimicrobial resistance occurs when pathogens resist exposure to antibiotics that are intended to kill or slow their growth [1,2]. It has become one of the leading public health threats of the 21st century, posing significant dangers to global health, economic stability, and the effectiveness of healthcare systems [3]. Globally, bacterial Antimicrobial Resistance(AMR) was estimated to be responsible for approximately 1.27 million deaths and associated with about 4.95 million deaths in 2019, making it comparable to major infectious diseases such as HIV/AIDS and Malaria in terms of mortality burden [4]. The World Health Organization (WHO) has recognised antimicrobial resistance as a serious public health concern that can result in prolonged hospital admissions, greater medical expenses and increased mortality rates [5]. The AWaRe framework guides rational antibiotic prescribing, promotes the use of first-line Access antibiotics when appropriate, restricts the use of Watch antibiotics due to their higher resistance potential, and reserves last-resort Reserve antibiotics for treatment of multidrug-resistant infections, thereby playing a critical role in slowing the emergence and spread of antimicrobial resistance [6]. Furthermore, in 2017, the WHO identified ESKAPE pathogens: Enterococcus faecium, Staphylococcus aureus, Klebsiella pneumoniae, Acinetobacter baumannii, Pseudomonas aeruginosa, and Enterobacter species as major global health threats due to their rising antibiotic resistance and potential to cause serious infections in humans [2]. This has made standard treatments less effective as bacterial pathogens change, making it more difficult to treat once-treatable infections. This phenomenon is particularly pronounced in low- and middle-income countries, where inappropriate antibiotic abuse is widespread, infectious disease burdens are high, and healthcare resources are limited [3,7].

In Tanzania, the prevalence of antibiotic resistance has been increasingly documented; however, comprehensive data remain sparse [8,9]. Available hospital and laboratory-based studies report high levels of resistance often exceeding 40–70% to commonly used antibiotics among key bacterial pathogens, including members of the ESKAPE group [10–12]. A variety of factors contribute to the rising rates of AMR in the country, including over-prescription of antibiotics, self-medication, and inadequate regulatory frameworks for antibiotic use [13–16]. Additionally, socio-economic factors such as personal awareness, attitudes, practices, poverty, lack of access to

healthcare services, and poor sanitation and hygiene practices exacerbate the situation [13]. The interplay of these factors creates an environment conducive to the emergence and spread of resistant strains of bacteria.

Despite the critical threat posed by resistant pathogens, systematic reviews focusing specifically on Tanzania's antibiotic resistance landscape are limited. For instance, a review on *Staphylococcus aureus* documented rising rates of Methicillin Resistance Staphylococcus aureus(MRSA) and non-susceptibility to common prescribed antibiotics, which does not cover multiple priority pathogens across human health sectors or consolidate national prevalence data [17]. A consolidated analysis capturing the overall prevalence across the country is still needed [18–20]. The absence of pooled national estimates limits evidence-based policymaking, hinders optimization of empirical treatment guidelines, and weakens monitoring of progress under the National Action Plan on AMR.

Therefore, this systematic review and meta-analysis aimed to estimate the pooled prevalence of antimicrobial resistance among bacterial pathogens in Tanzania from 2014 to 2024 and evaluate resistance patterns across antibiotic classes using subgroup analyses by synthesising evidence from published studies. This review will inform the antimicrobial stewardship programmes, strengthen AMR surveillance, guide updates to treatment guidelines, and support the implementation of Tanzania's National Action Plan on Antimicrobial Resistance 2023–2028.

## Materials and methods

### Protocol registration

The systematic review and meta-analysis protocol was registered with PROSPERO under the ID number PROSPERO 2024 CRD42024608537.

### Settings

This review included studies conducted in Tanzania, an East African country located in the African Great Lakes Region. Tanzania is bordered by Uganda to the northwest, Kenya to the northeast, the Indian Ocean to the east, Mozambique and Malawi to the south, Zambia to the southwest, and Rwanda, Burundi, and the Democratic Republic of the Congo to the west. According to the 2022 national census, Tanzania has a population of approximately 62 million [21], making it the most populous country south of the equator.

### Review procedures

The systematic review process was initiated by formulating the research question using the PICOS (Population, Intervention, Comparison, Outcomes, and Study Design) framework. The population included all humans regardless of age and sex. Since the focus was on prevalence, no specific intervention or comparison was required. The primary outcome was the reported prevalence of Antimicrobial Resistance in Tanzania. The review process was conducted and finalized in accordance with the Preferred Reporting Items for Systematic Reviews and Meta-Analysis (PRISMA) guidelines, which guided the identification, analysis, and reporting of the findings across all sections of the manuscript, including the title, abstract, introduction, methods, results and discussion [22].

### Search strategy

Relevant published studies were searched using electronic databases such as PubMed, ScienceDirect, and Google Scholar. The literature search was conducted on 18th October 2024. The following phrases were used for the search: (((("Prevalence"[Mesh] OR "Prevalences" OR "Point Prevalence" OR "Point Prevalences" OR "Prevalence, Point" OR "Period Prevalence" OR "Period Prevalences" OR "Prevalence, Period") AND ("Risk Factors"[Mesh] OR "Factor, Risk" OR "Risk Factor" OR "Population at Risk" OR "Populations at Risk" OR "Risk Scores" OR "Risk Score" OR "Score, Risk" OR "Risk Factor Scores" OR "Risk Factor Score" OR "Score, Risk Factor" OR "Health Correlates" OR "Correlates, Health"

OR "Social Risk Factors" OR "Factors, Social Risk" OR "Risk Factor, Social" OR "Risk Factors, Social" OR "Social Risk Factor")) AND ("Drug Resistance, Bacterial"[Mesh]OR "Antibacterial Drug Resistance" OR "Antibiotic Resistance, Bacterial")) AND ("Tanzania"[Mesh] OR "United Republic of Tanzania" OR "Zanzibar").

**Selection criteria and search outcomes**

The following criteria were used to select relevant studies: (i) studies reporting on the prevalence of antibiotic resistance in bacterial pathogens; (ii) studies conducted in Tanzania over ten years, from 2014 to 2024; (iii) original cross-sectional studies, cohort studies, case-control studies, experimental studies and modelling studies. The review excluded studies that did not focus on antibiotic resistance, animal or laboratory-based studies without clinical relevance, reviews, letters, notes, editorials, and conference reports. This process yielded 18,265 articles related to the study topic. For Google Scholar, results were sorted by relevance, and the first 160 pages (1600 results) were screened. The studies were uploaded to Covidence, a web-based software for screening, full article review, and data extraction [23]. A total of 28 studies satisfied the set criteria and were included in this review (**Fig 1**).

**Data synthesis, analysis and reporting**

Data were independently extracted by two reviewers using a data extraction form created in Covidence. The form captured information on the author and year of publication, study setting, design, population, sample size, bacterial isolates, specimen sources, antibiotics tested, and resistance prevalence. Discrepancies between reviewers were resolved through discussion, and when consensus was not reached, a third reviewer adjudicated. The methodological quality of the included studies was independently assessed by two reviewers using the JBI Critical Appraisal Checklist for Prevalence Studies, and any discrepancies were resolved through consensus. Descriptive statistics and meta-analysis were conducted using R version 4.4.1. Frequencies and percentages were used to summarize study characteristics, and a random effect model was applied to estimate pooled prevalence rates of antibiotic resistance. $I^2$ statistics were used to evaluate heterogeneity among the studies, with values of 75% or higher considered indicative of significant heterogeneity [24]. Subgroup analyses were conducted to investigate potential drug resistance in specific pathogens associated with each antibiotic. All analyses were conducted using the R libraries (meta) and (metafor) [25,26]. Finally, the results were visually presented through tables and figures, including bar charts and plot forests.

**Quality assessment**

The included studies were evaluated for methodological quality and risk of bias using the Joanna Briggs Institute#39;s Quality Assessment Tool for Cross-Sectional Studies [27,28]. The studies were scored as (yes = 1; no or unclear = 0). Independent reviewers recorded judgments and supporting evidence within Covidence, and any disagreements were resolved through consensus. The analysis showed that the included studies had a low risk of bias, with an average total score of 75%, which exceeds the 50% threshold defined for acceptable quality (**S1 Table**).

**Classification of resistance levels**

Antimicrobial resistance was classified based on the proportion of resistant isolates reported in the included studies. Resistance was categorized as high when the prevalence of resistance was ≥ 50% and low when the prevalence was < 50%. This classification approach has been used in previous antimicrobial resistance surveillance studies and aligns with commonly applied thresholds in epidemiological analyses to facilitate comparison across settings and pathogens [29].

**Ethical considerations**

As this study used published data only, institutional review board approval was not required

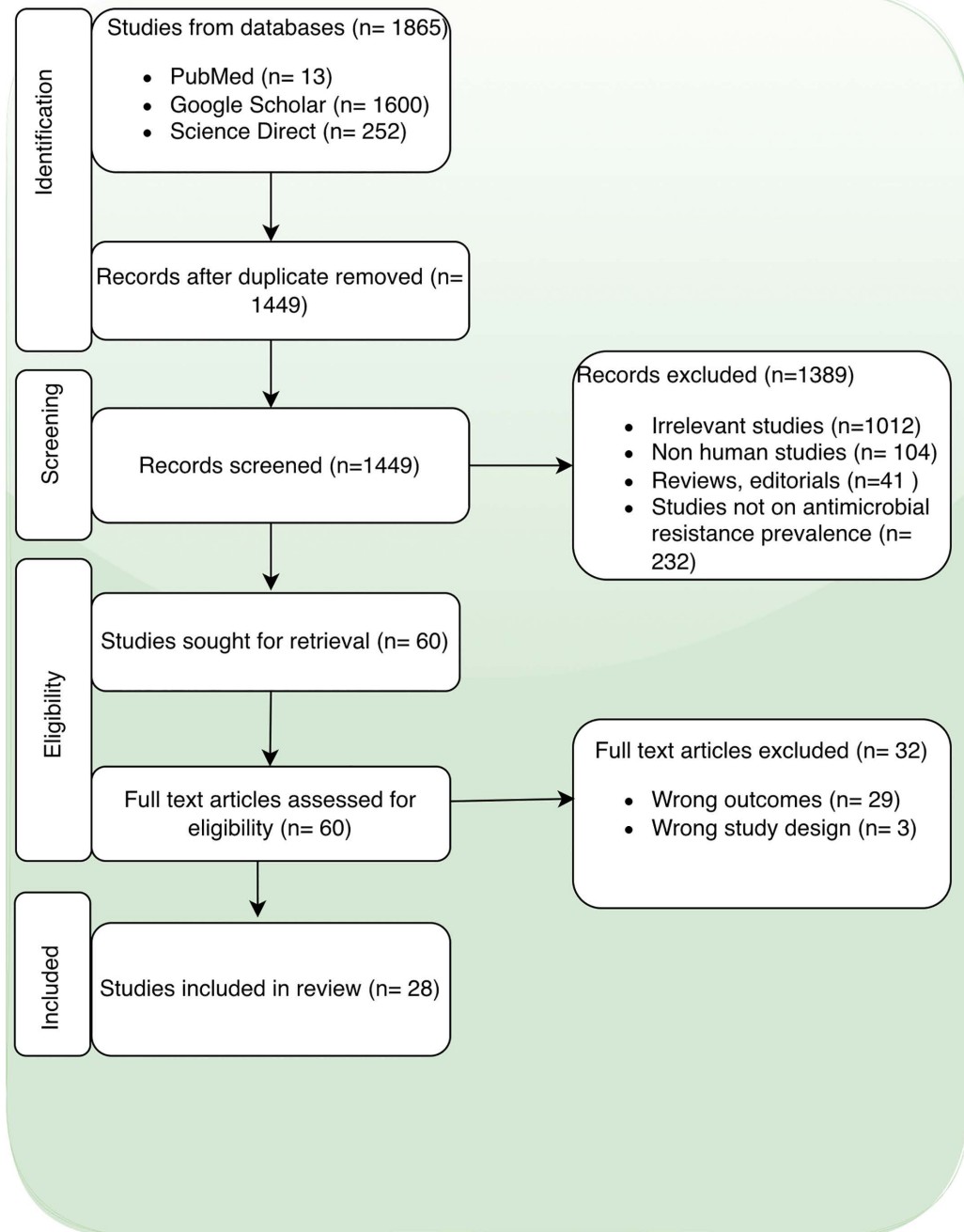

**Fig 1. Prisma flow chart of the systematic review and article selection.**

## Results

### Description of included studies

**Settings, study designs and year of publication of the selected studies.** Of the 28 included studies, 9(32%) were conducted in Dar es Salaam, followed by Mwanza 7(25%) and Kilimanjaro 5(17.9%), with 75.0% being cross-sectional. Additionally, 17.9% of the studies were published in 2020, while 14.3% were published in 2016, 2019, and 2022 (**Fig 2**).

**Study population, sample size and number of isolates.** Most of the selected studies (46%) involved children under five years of age. The sample size of the selected studies ranged from 75 [22] to 4,306 [23], with the number of isolates ranging from 22 [24] to 4,030 [23] (**Table 1**). The most common clinical samples were urine (25%), nasopharyngeal swabs (10.7%), open wound pus swabs (7.1%), and blood (3.6%) (**Fig 3**).

## Results from meta-analysis

The meta-analysis included 28 studies that investigated antibiotic resistance patterns across a broad spectrum of pathogens, including ESKAPE-E bacteria (*Escherichia coli, Staphylococcus aureus, Klebsiella pneumoniae, Acinetobacter baumannii, Pseudomonas aeruginosa,* and *Enterobacter spp.*), *Proteus spp., Streptococcus spp., Campylobacter spp.,* and *P. mirabilis.*

The review examined the resistance of various pathogens against 17 antibiotics, classified according to the WHO AWaRe framework [6]. *Access group* antibiotics included penicillin (*amoxicillin* and *ampicillin*), macrolides (erythromycin), tetracyclines (*tetracycline*), and nitrofuran compounds (*nitrofurantoin*). *Watch group* antibiotics comprised fluoroquinolones (*ciprofloxacin*), sulfonamides (*trimethoprim-sulfamethoxazole*), and cephalosporins (*ceftriaxone, ceftazidime, cefotaxime, cefotaxin* and *cefuroxime*). *Reserve group* antibiotics included carbapenems (*meropenem* and *imipenem*), aminoglycosides (*amikacin*), lincosamides (*clindamycin*), and cephalosporins (*cefepime*) (**S1–S14 Figs in S1 File**) (**Fig 4–6**).

A subgroup analysis was conducted to assess the resistance patterns within each antibiotic class for specific pathogens. The results revealed high resistance to penicillin (*amoxicillin–clavulanic acid*), with the highest pooled prevalence

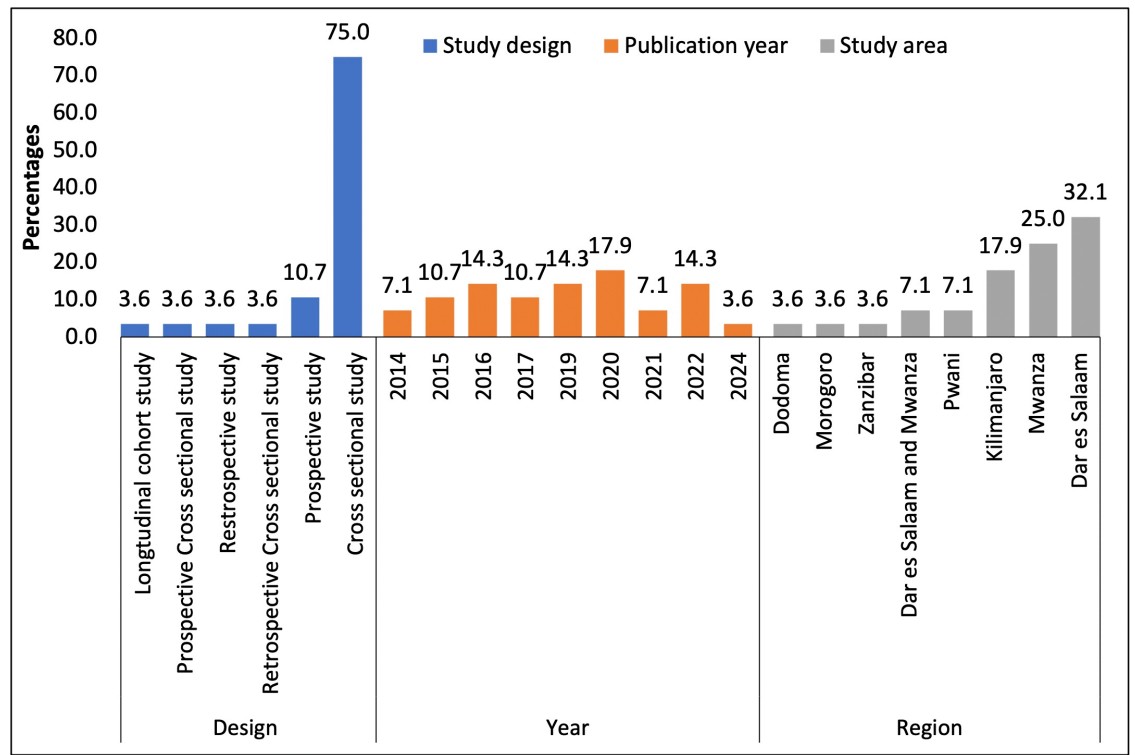

**Fig 2. Distribution of the Year of Publication, Study Design, and Region.**

**Table 1. Authors, study population, sample size and number of isolates of the selected studies.**

| Author | Study population | Sample Size | Number of Isolates |
|---|---|---|---|
| Manyahi et al., 2014 [30] | Patients with clinical evidence of Surgical Site Infection | 100 | 147 |
| Moyo et al., 2014 [31] | Healthy under-fives | 114 | 114 |
| Marwa et al., 2015 [32] | HIV/AIDS and Non HIV/AIDS patients | 945 | 155 |
| Komba et al., 2015 [33] | Patients with enteric and non-enteric symptoms | 1195 | 136 |
| Ahmed et al., 2015 [34] | Malnourished under-fives | 402 | 84 |
| Okamo et al., 2016 [35] | Medical students | 314 | 66 |
| Moremi et al., 2016 [36] | More samples were from paediatric population | 3330 | 439 |
| Tellevik et al., 2016 [37] | Children Below 2 years | 603 | 284 |
| Seidman et al., 2016 [38] | Young children | 377 | 2492 |
| Ahmed et al., 2017 [34] | Malnourished under-fives | 402 | 42 |
| Joachim et al., 2017 [39] | Patients attended at Emergency Department /Inpatient Department | 258 | 22 |
| Kumburu et al., 2017 [40] | Patient admitted in surgical and medical department | 575 | 377 |
| Kiponza et al., 2019 [41] | Post-delivery women | 197 | 107 |
| Seni et al., 2019 [42] | Pregnant women with significant bacteriuria | 1828 | 323 |
| Emgård et al., 2019 [43] | Children | 775 | 244 |
| Madut et al., 2019 [44] | Adolescent and adult patients aged ≥13 years with fever | 1648 | 31 |
| Daud et al., 2020 [45] | Children | 282 | 282 |
| Kamgobe et al., 2020 [46] | Pregnant women with or without Premature Rupture of the Membrane | 350 | 39 |
| Mikomangwa et al., 2020 [47] | Patients attended at Muhimbili National Hospital | 183 | 201 |
| Manyahi et al., 2020 [48] | All inpatients with features suggestive of blood stream infection | 402 | 44 |
| Kadigi et al., 2020 [49] | Under-fives with symptoms of UTI | 270 | 80 |
| Letara et al., 2021 [50] | Children | 350 | 76 |
| Sangeda et al., 2021 [51] | Under-fives with symptoms of UTI | 214 | 35 |
| Manyahi et al., 2022 [52] | Admitted patients | 198 | 179 |
| Masoud et al., 2022 [12] | Admitted patients | 75 | 76 |
| Schmider et al., 2022 [53] | Outpatients aged ≥2 years with symptoms of UTI | 270 | 119 |
| Silago et al., 2022 | Patients who resided within the study area | 1327 | 412 |
| Muhiddin Hamada Omar 2024 [54] | Hospitalized patients | 4306 | 4030 |

observed in *Acinetobacter baumannii* (0.92 [0.76–0.98]), followed by *Proteus mirabilis* (0.87 [0.63–0.96]), *Klebsiella pneumoniae* (0.85 [0.73–0.92]), and *Enterobacter spp.* (0.83 [0.61–0.94]). Resistance was also notable among *Escherichia coli* (0.61 [0.42–0.77]) and *Klebsiella spp.* (0.53 [0.20–0.84]) (**Fig 4**). Similarly, a high level of resistance to penicillin (ampicillin) was observed among ESKAPE-E pathogens and Proteus spp. The pooled prevalence of resistance ranged from Klebsiella pneumoniae (0.96 [0.83–0.99]), Acinetobacter baumannii (0.94 [0.67–0.99]), Escherichia coli (0.90 [0.81–0.95]), Klebsiella spp. (0.90 [0.81–0.95]), Enterobacter cloacae (0.78 [0.46–0.94]), Pseudomonas aeruginosa (0.77 [0.47–0.92]), Proteus spp. (0.76 [0.58–0.88]), and Staphylococcus aureus (0.76 [0.38–0.94]) (**S1 Fig in** S1 File).

Resistance to macrolides, specifically erythromycin, was highest among *Campylobacter spp.* (0.85 [0.80–0.89]), followed by *Escherichia coli* (0.61 [0.16–0.93]) and *Staphylococcus aureus* (0.52 [0.35–0.68]) (**S2 Fig in** S1 File). Ciprofloxacin exhibited a pooled resistance prevalence of 0.54 [0.33–0.73] in *Acinetobacter baumannii* (**S3 Fig in** S1 File), whereas amikacin resistance was notably high in *Proteus spp.* (0.86 [0.35–0.99]) (**S4 Fig in** S1 File). Tetracycline resistance was prevalent in *Klebsiella pneumoniae* (0.81 [0.24–0.98]) and *Escherichia coli* (0.77 [0.71–0.83]) (**S5 Fig in** S1 File). Additionally, trimethoprim-sulfamethoxazole showed high resistance in *Klebsiella pneumoniae* (0.93 [0.77–0.98]), *Escherichia coli* (0.86 [0.75–0.93]), and *Proteus mirabilis* (0.80 [0.30–0.97]) (**S6 Fig in** S1 File).

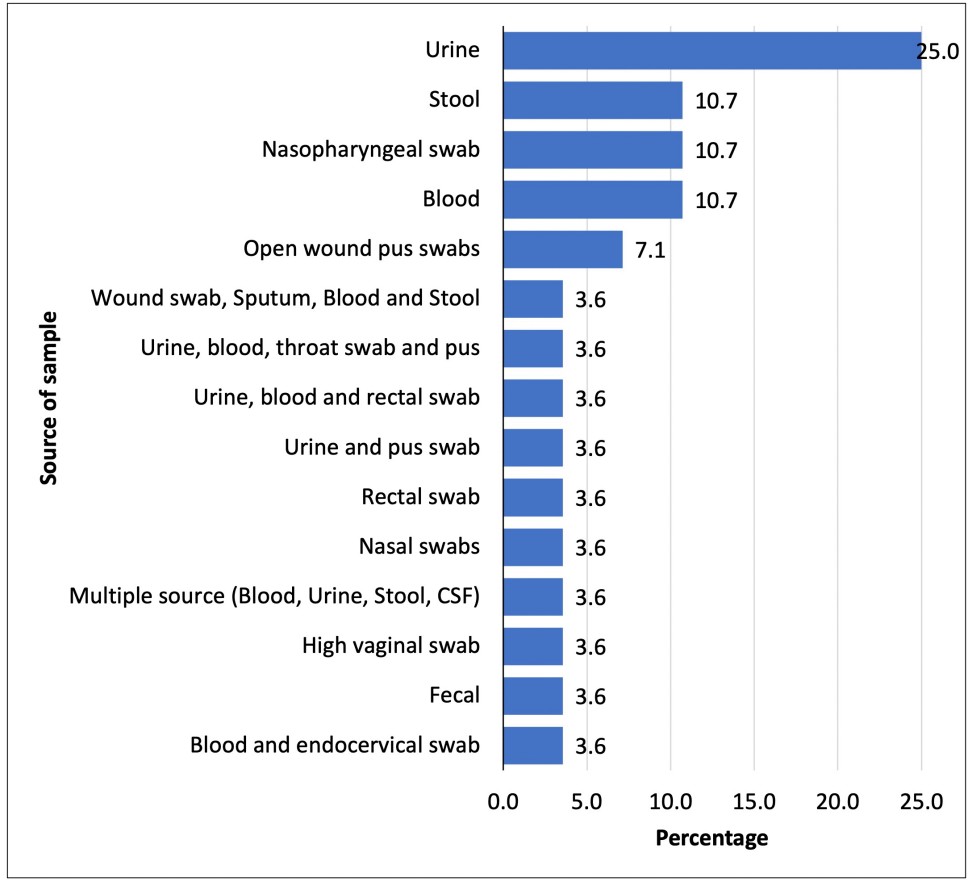

**Fig 3. Sources of Samples for Selected Studies.**

The pathogens exhibiting the highest resistance to ceftriaxone, a third-generation cephalosporin, include Acinetobacter baumannii (0.91 [0.70–0.98]), Proteus mirabilis (0.71 [0.47–0.87]), Klebsiella species (0.65 [0.40–0.84]), Klebsiella pneumoniae (0.74 [0.59–0.85]), and Pseudomonas aeruginosa (0.85 [0.74–0.92]) (**Fig 5**). Similarly, high resistance to cefotaxime was observed in Acinetobacter baumannii (0.92 [0.53–0.99]), Pseudomonas aeruginosa (0.90 [0.53–0.99]), Klebsiella pneumoniae (0.94 [0.62–0.99]), Escherichia coli (0.81 [0.35–0.97]), and Proteus mirabilis (0.81 [0.24–0.98]) (**S7 Fig in S1 File**).

For ceftazidime, the highest resistance was noted among Klebsiella pneumoniae (0.89 [0.64–0.97]), Acinetobacter baumannii (0.82 [0.58–0.93]), and Proteus mirabilis (0.67 [0.44–0.84]) (**S8 Fig in S1 File**). Resistance to the fourth-generation cephalosporin, cefepime, was also observed across multiple bacterial species, with Enterobacter species (0.91 [0.66–0.98]), Escherichia coli (0.86 [0.14–1.00]), Klebsiella species (0.73 [0.59–0.83]), and Citrobacter species (0.72 [0.47–0.88]) (**S9 Fig in S1 File**).

Despite the overall high resistance observed among pathogens against specific antibiotics, certain pathogens exhibited a lower pooled resistance prevalence. Escherichia coli and Staphylococcus aureus showed resistance to cefuroxime (0.38 [0.19–0.60]) and cefoxitin (0.38 [0.26–0.53]), respectively (**S10–S11 Figs in S1 File**). The ESKAPE-E pathogens were tested against meropenem, exhibiting an overall pooled resistance prevalence of 0.11 [0.06–0.19]). Among individual pathogens, Escherichia coli showed the lowest resistance (0.04 [0.01–0.10]), followed by Klebsiella species (0.07

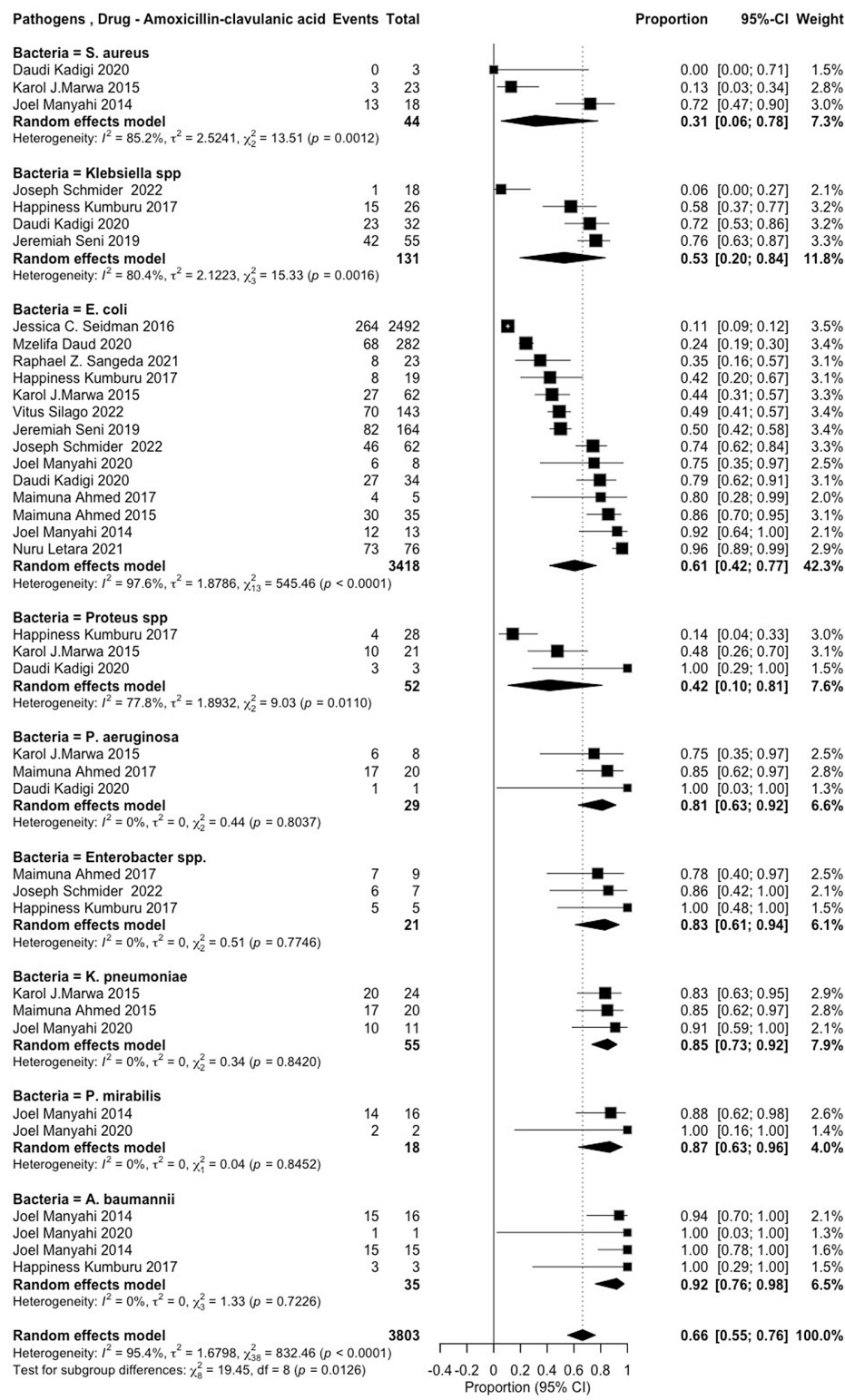

**Fig 4. Amoxicillin/clavulanic acid resistance patterns among various pathogens.**

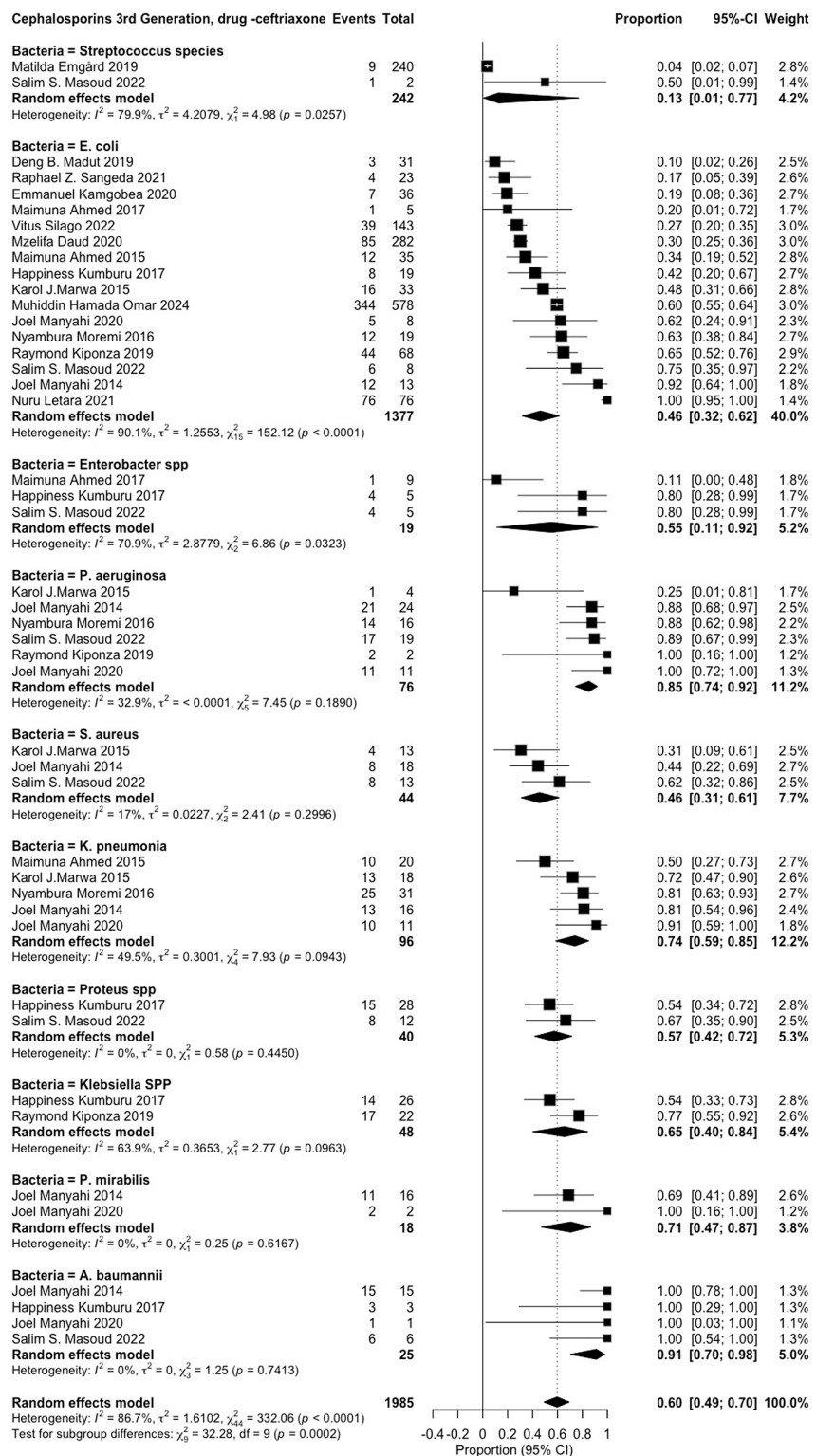

**Fig 5. Ceftriaxone resistance patterns in various pathogens.**

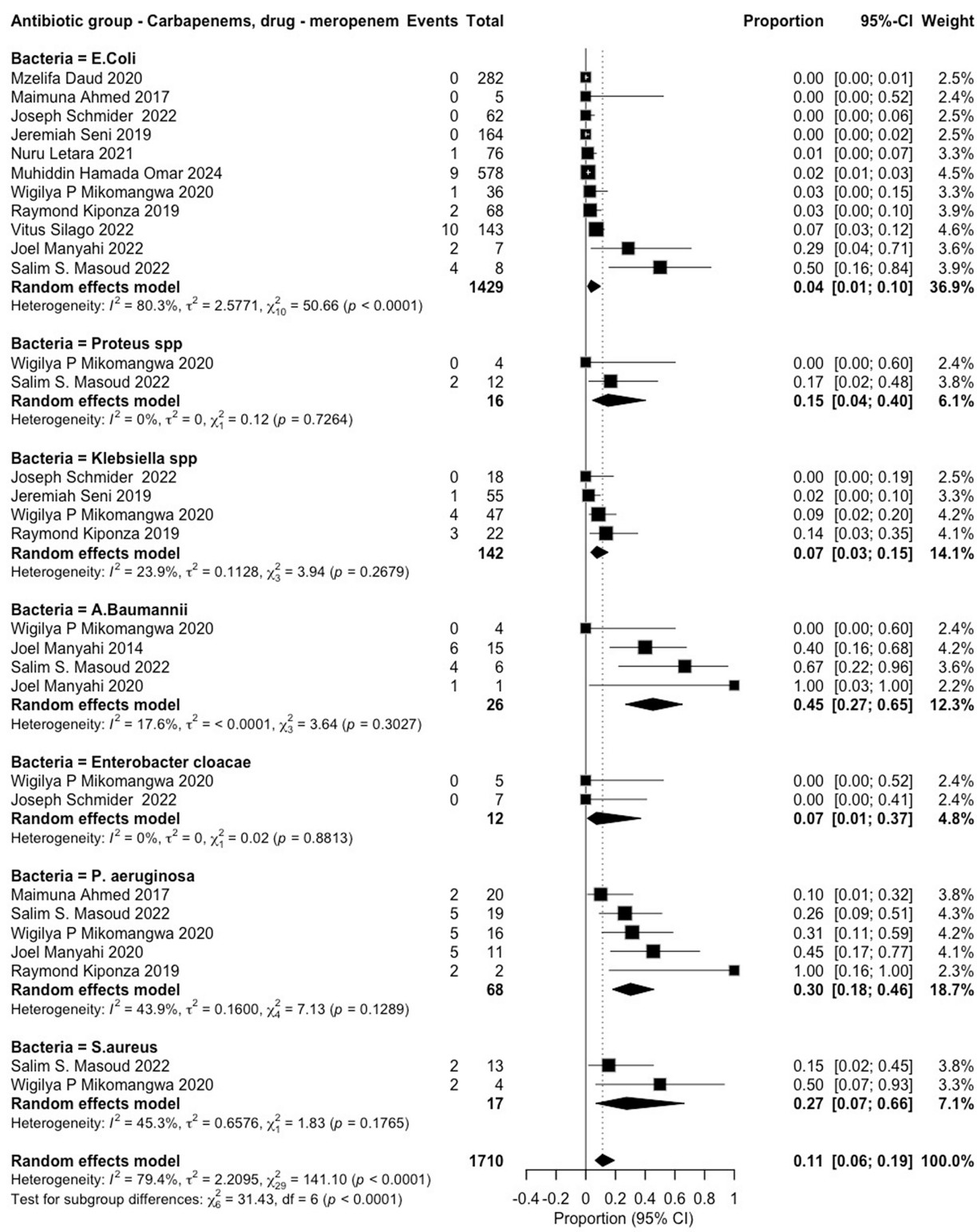

**Fig 6. Meropenem resistance patterns among various pathogens.**

[0.03–0.15]), Enterobacter cloacae (0.07 [0.01–0.37]), Pseudomonas aeruginosa (0.30 [0.18–0.46]), and Staphylococcus aureus (0.27 [0.07–0.66]) (**Fig 6**). Notably, Acinetobacter baumannii exhibited the highest resistance (0.45 [0.27–0.65]). For imipenem, Klebsiella pneumoniae and Escherichia coli had an overall pooled resistance prevalence of 0.06 [0.02–0.26], with individual resistance rates of (0.08 [0.02–0.26]) and (0.06 [0.02–0.26]), respectively (**S12 Fig in** S1 File). Similarly, Klebsiella pneumoniae and Escherichia coli exhibited an overall pooled resistance prevalence of (0.06 [0.02–0.14]) to clindamycin (**S13 Fig in** S1 File).

In the case of nitrofurantoin, Staphylococcus aureus, Escherichia coli, and Klebsiella species demonstrated an overall pooled resistance prevalence of (0.24 [0.16–0.34]) (**S14 Fig in** S1 File). The included studies demonstrated moderate to substantial heterogeneity across pathogens and antibiotics, with 76.5% exhibiting high heterogeneity (above 75%).

## Discussion

This systematic review examined the prevalence of antimicrobial resistance in Tanzania and revealed concerning levels of resistance across antibiotics classified under the WHO AWaRe framework. Clinically important pathogens, including ESKAPE-E, demonstrated widespread resistance, with particularly high resistance rates observed against the commonly used *Access group* of antibiotics, including *nitrofurantoin, erythromycin, amoxicillin-clavulanic acid, Tetracycline, and ampicillin.* Comparable resistance patterns have been documented in South Africa and across East African countries, including Uganda, Kenya, Ethiopia, Rwanda, and the Democratic Republic of Congo [55,56]. These elevated resistance levels significantly compromise treatment efficacy, leading to prolonged illness, increased healthcare costs, and higher mortality rates [8,57].

Considering individual pathogens and specific *Access group* of antibiotics, resistance to *ampicillin* was particularly alarming among Klebsiella pneumoniae, Acinetobacter baumannii, and Escherichia coli isolates. This resistance is likely driven by widespread β-lactamase production and the prolonged empirical use of penicillin in both community and hospital settings, which promotes selective pressure on these pathogens [58–60]. Given that ampicillin is recommended as a treatment option for these pathogens in the Tanzania Standard Treatment Guidelines [61], this high level of resistance raises concerns about treatment complications and prolonged hospital stays. These findings align with previous studies conducted in South Africa and East African countries [55,56], suggesting that the challenge is not isolated to Tanzania but reflects a broader regional trend, thereby underscoring the urgent need for targeted interventions. Similarly, Acinetobacter baumannii, Proteus mirabilis, and Klebsiella pneumoniae demonstrated substantial resistance to *amoxicillin-clavulanic acid*, complicating the management of diverse infections [30,62–64]. In contrast, nitrofurantoin exhibited comparatively low resistance among *E.coli, S. aureus* and *Klebsiella* species, supporting its continued use for UTIs [61]. This low resistance is pharmacologically explained by its multifaceted mechanism of action, which targets multiple bacterial enzymes, and its limited systemic absorption, both of which minimize selective pressure and reduce the likelihood of resistance development [65]. Notably, erythromycin resistance was also observed, particularly in *Campylobacter* species, *E. coli*, and *S. aureus*, driven by specific genetic mechanisms [66]. This pattern, consistent with trends seen in Kenya and across Africa [67,68], underscores growing concerns over the waning efficacy of erythromycin.

The *Watch group* of antibiotics exhibited notable resistance to *sulfonamides, ceftriaxone, ceftazidime, cefuroxime,* and *ciprofloxacin.* Ceftriaxone, a crucial antibiotic for severe bloodstream infections, surgical site infections, complicated UTIs, and gastrointestinal infections, demonstrated particularly high resistance across multiple pathogens, including A. baumannii, Pseudomonas aeruginosa and K. pneumoniae [69]. This resistance is likely driven by the acquisition of resistance genes, such as beta-lactamases, efflux pumps, and other adaptive mechanisms that reduce antibiotic susceptibility [70,71]. These findings are consistent with a 2019 review conducted focusing on A. baumannii, which reported similar elevated levels, underscoring the limited efficacy of ceftriaxone in managing severe bacterial infections [60]. However, Ciprofloxacin resistance was also observed among ESKAPE-E pathogens, with *A. baumannii* showing particularly high resistance, while other pathogens exhibited comparatively lower levels. This high resistance in A. *baumannii* may

indicate selective pressure due to the widespread use of ciprofloxacin in post-operative care and bloodstream infections [54,61,62]. Biologically, this resistance is mediated by mutations in DNA gyrase and topoisomerase IV, as well as overexpression of efflux pumps that actively remove the drug from bacterial cells [72,73].

The *Reserve group* of antibiotics, designated for use when alternative treatment options have proven ineffective, including *carbapenems*, *amikacin*, *clindamycin*, and *cefepime*, exhibited variable resistance levels ranging from 6% to 81%. This trend raises concerns about treatment outcomes for complicated cases referred from lower-level facilities, where *Access group* antibiotics have failed. When analysing resistance by specific antibiotic-pathogen combinations, both *clindamycin* and *imipenem* exhibited relatively low resistance rates against E. coli and K. pneumoniae, respectively. The lower resistance observed for these antibiotics can be attributed to restricted access, as they are not available over the counter, thereby reducing misuse and selective pressure [74]. These findings align with studies conducted in Uganda and South Africa, which also reported lower resistance rates to these reserved antibiotics [75,76].

Among ESKAPE-E pathogens, *meropenem* demonstrated relatively low levels of resistance among E. coli, Klebsiella species, and Enterobacter cloacae. However, A. baumannii exhibited comparatively higher resistance, which may reflect its well-documented capacity to rapidly develop multidrug resistance [77]. These findings underscore the importance of maintaining strict stewardship of *reserve* antibiotics while strengthening continuous surveillance to prevent further escalation of resistance. Similar trends have been observed in studies conducted in South Africa and Ethiopia [75,78], highlighting regional similarities in antimicrobial resistance trends and reinforcing the need for coordinated and sustained monitoring efforts across similar settings. In contrast, cefepime and amikacin showed higher resistance among several pathogens, particularly *E. coli*, with notable resistance also observed among *Enterobacter* species and *Proteus* species, suggesting that some antibiotics traditionally considered reserved options may be experiencing reduced effectiveness.

Despite these important findings, several limitations should be considered when interpreting the results. High heterogeneity was observed across most pooled estimates, likely due to differences in study design, year of publication, study populations, laboratory methods, and antimicrobial susceptibility testing practices, which may affect the precision of the pooled prevalence estimates. Most included studies were hospital-based and conducted in urban settings, particularly in Dar es Salaam and Mwanza, limiting generalizability to rural and primary healthcare facilities, which were underrepresented. Variation in laboratory testing standards across studies may also have influenced the comparability of resistance patterns. In addition, potential publication bias cannot be ruled out, as unpublished studies and routine surveillance data may not have been captured; formal assessment of publication bias was not performed due to methodological limitations, as standard tests such as Egger's regression test are often underpowered or inappropriate for prevalence meta-analyses. Finally, temporal trend analysis was not feasible because of inconsistent reporting periods across studies.

## Conclusion

The current review highlights the critical and troubling rates of antimicrobial resistance in Tanzania, which create significant challenges in the effective management of urinary tract infections, upper respiratory tract infections, skin infections, bloodstream infections, pneumonia, and wound infections. The high resistance to the *Access* and *Watch* groups of antibiotics indicates that standard treatment regimens may no longer be reliable, resulting in longer illnesses, increased healthcare expenses, and higher mortality rates. Although resistance levels to reserved antibiotics like *carbapenems, cefepime* and *amikacin* are relatively low, troubling trends have been observed, especially with *A. baumannii, E. coli, Proteus* species and *Enterobacter* species. This situation emphasises the need to safeguard Reserve antibiotics through strengthened antimicrobial stewardship programmes, robust regulatory policies, and periodic revision of the Tanzania Standard Treatment Guidelines. Furthermore, enforcing prescription regulations is crucial in mitigating the ongoing emergence and spread of antimicrobial resistance.

## Supporting information

**S1 Table. Quality Assessment Table.**
(XLSX)

**S1 File. Antibiotic resistance patterns among bacterial pathogens.**
(ZIP)

**S1 Fig. PRISMA 2020 Checklist.**
(DOCX)

## Acknowledgments

The authors would like to thank the researchers who published their articles online, without whom this review would not have been possible.

## Author contributions

**Conceptualization:** Charles Basil Kafaiya, Johnson Mshiu, Obadia Bishoge, Irene Mremi, Mwanaada Kilima, Sia Malekia.

**Data curation:** Charles Basil Kafaiya, Johnson Mshiu, Irene Mremi.

**Formal analysis:** Johnson Mshiu.

**Methodology:** Charles Basil Kafaiya, Johnson Mshiu, Angelina M. Lutambi, Obadia Bishoge, Irene Mremi, Sia Malekia.

**Project administration:** Jonathan Mcharo, Mary Mayige, Said Aboud.

**Resources:** Charles Basil Kafaiya, Jonathan Mcharo, Mary Mayige, Said Aboud.

**Software:** Charles Basil Kafaiya.

**Supervision:** Angelina M. Lutambi, Obadia Bishoge, Jonathan Mcharo, Mary Mayige, Said Aboud.

**Validation:** Charles Basil Kafaiya, Angelina M. Lutambi, Irene Mremi, Mwanaada Kilima, Mary Mayige.

**Visualization:** Charles Basil Kafaiya, Angelina M. Lutambi, Obadia Bishoge, Irene Mremi, Mwanaada Kilima, Sia Malekia.

**Writing – original draft:** Charles Basil Kafaiya, Johnson Mshiu, Angelina M. Lutambi, Obadia Bishoge, Irene Mremi, Mwanaada Kilima, Sia Malekia.

**Writing – review & editing:** Charles Basil Kafaiya, Johnson Mshiu, Angelina M. Lutambi, Obadia Bishoge, Irene Mremi, Mwanaada Kilima, Sia Malekia, Jonathan Mcharo, Mary Mayige, Said Aboud.

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
