## [Decision Letter · Decision Letter 0]

26 Dec 2025

Dear Dr. Kafaiya,

Thank you for submitting your manuscript to PLOS ONE. After careful consideration, we feel that it has merit but does not fully meet PLOS ONE’s publication criteria as it currently stands. Therefore, we invite you to submit a revised version of the manuscript that addresses the points raised during the review process.

We look forward to receiving your revised manuscript.

Kind regards,

Mabel Kamweli Aworh, DVM, MPH, PhD. FCVSN

Academic Editor

PLOS One

Journal Requirements:

https://journals.plos.org/plosone/s/file?id=ba62/PLOSOne_formatting_sample_title_authors_affiliations.pdexf

2. Thank you for submitting your work to PLOS ONE. PLOS' guidelines require systematic reviews to be comprehensive and therefore up-to-date (https://journals.plos.org/plosone/s/submission-guidelines#loc-systematic-reviews-and-meta-analyses). Specifically, we expect scoping review searches not to have been done earlier than one year before the time of submission. To this effect, please include in the Methods section the date of the most recent search of the literature review conducted for your study. Thank you for your attention to this request.

4. Please remove all personal information, ensure that the data shared are in accordance with participant consent, and re-upload a fully anonymized data set.

Additional Editor Comments:

In addition to addressing all reviewer comments, please ensure that all scientific names are italicized throughout the manuscript, including the abstract, main text, figures, and tables. For example, in the abstract (lines 30–39), *Klebsiella pneumoniae*, , , , *Acinetobacter baumannii*, and , and , and , and *Escherichia coli* should be italicized.should be italicized.should be italicized.should be italicized.

Reviewers' comments:

Reviewer's Responses to Questions

**Comments to the Author**

1. Is the manuscript technically sound, and do the data support the conclusions?

Reviewer #1: Partly

Reviewer #2: Yes

Reviewer #3: Yes

Reviewer #4: Partly

2. Has the statistical analysis been performed appropriately and rigorously?

Reviewer #1: Yes

Reviewer #2: Yes

Reviewer #3: Yes

Reviewer #4: N/A

3. Have the authors made all data underlying the findings in their manuscript fully available?

Reviewer #1: Yes

Reviewer #2: Yes

Reviewer #3: Yes

Reviewer #4: No

4. Is the manuscript presented in an intelligible fashion and written in standard English?

Reviewer #1: Yes

Reviewer #2: Yes

Reviewer #3: Yes

Reviewer #4: Yes

Reviewer #1: I wish to thank editor for asking me to review this manuscript on prevalence of AMR in Tanzania.

Comments

General and abstract section

Authors aimed to perform the systematic review and meta-analysis to present prevalence of AMR in Tanzania. Authors may clarify if they included animal and environmental AMR studies and if not why they are excluded (just to confirm).

Results section

Authors included ESKAPE-E bacterial organisms of which majority are multidrug resistant. All except Streptococcus spp. are intrinsically resistant to penicillin/ampicillin. These organisms are mostly acquired in healthcare settings and the most of access antibiotics from AWaRe classes are rather ineffective. They may apply to community infections.

KP as intrinsically resistant to penicillin/ampicillin should be removed from analysis and the same for A. baumannii and other from the group. Klebsiella pneumoniae exhibits intrinsic resistance to ampicillin due to the presence of the SHV-1 penicillinase gene on its chromosome. The antibiotic resistome concept is predicated on the understanding that soil-dwelling environmental bacteria are producers of antimicrobial compounds. Thus, many components of the resistome must have evolved long before the use of antibiotics became clinically common. Based on knowledge about the bacteria in the group one may suggest to reanalyze data and focus on watch and reserve antibiotics and to categorize according settings.

End of comments

Reviewer #2: The review notes unclear objectives, limited clarity on AMR terms and AWaRe, methodological gaps, data inconsistencies, and unaddressed limitations, though it effectively synthesizes resistance trends to inform policy.

Please view additional comments in the attachment.

Reviewer #3: The manuscript was well written in English and followed the PRISMA guideline which ensures technicality. Below are some concerns and recommendation:

Abstract

1. Please include justification for conducting the study in the abstract

2. Adjust the statement “Heterogeneity was assessed using forest plots”. A forest plot is not used to assess heterogeneity. Heterogeneity is assessed using I² statistic.

3. Is the prevalence of antimicrobial resistance defined by descriptive or meta-analysis?

4. Adjust the grammar in this statement: “Resistance to meropenem was lowest among Escherichia coli (0.04 [0.01–0.10]) and Klebsiella spp. (0.07 [0.03–0.15]), with an overall pooled 37 resistance to the ESKAPE-E pathogen of (0.11[0.06–0.19]).”

5. Specify who this recommendation is targeted at “The variability in resistance patterns underscore the need to re-evaluate empirical treatment protocols (STG/NEMLIT) to ensure effective treatment regimens, strengthening antimicrobial stewardship, enhancing surveillance systems, and promote rational antibiotic use.”

Introduction

6. Line 47 – 59: Can you quantify the burden of antimicrobial resistance globally and in Tanzania, respectively?

7. Clarify the language in this statement: “This phenomenon is particularly noticeable in low- and middle-income countries, where antibiotic abuse is pervasive, infectious disease burdens are high, yet scarce healthcare resources”

8. Provide references for the statement: “A variety of factors contribute to the rising rates of Antimicrobial Resistance (AMR) in the country, including over-prescription of antibiotics, self-medication, and inadequate regulatory frameworks for antibiotic use.”

9. What do you mean by “urgency” in line 68?

10. Line 69 -70: describe and cite what other reviews have done or did not do. Also, cite some individual studies on antimicrobial resistance in Tanzania that you referred to in this section.

11. Address the grammar in this statement: While several studies have reported resistance patterns for specific pathogens, therefore a consolidated analysis that captures the overall prevalence across the country was needed

Method

12. I think the information on lines 91 – 98 should be part of your introduction

13. State what approach was used to define the research question. If you used the PICO approach, include it in your “Review procedures section.”

14. Justify why you focused only on studies across 10 years

15. In your selection criteria, is there a difference between “studies reporting on the prevalence of antibiotic resistance in bacterial pathogens” and “studies focused on antibiotic resistance”

16. Did you predefine the bacterial pathogens and antibiotics analyzed in this review? If yes, you should justify why they were selected

17. State how many reviewers conducted the quality assessment?

18. Include the statistical package you used to conduct descriptive statistics.

19. Usually, only the I² statistic is used to evaluate heterogeneity and not the forest plot.

20. What cut-off was used to determine the higher value for the I² statistic?

Results

21. More specific numbers should be used instead of percentages to report on the number of studies. When using percentages, maintain consistency in reporting. Also, ensure that the papers conducted at each location are cited.

22. Line 186 – 190. You mentioned 17 antibiotics but only listed 10. You should include the other seven. You should define the criteria that were used to categorize resistance as high or low.

Discussion

23. This is new; it was not mentioned in the result. Ensure that you discuss only the results presented.

24. I suggest you focus on interpreting the results rather than repeating them in the discussion section.

25. Line 306: “Reserved group.” This was not mentioned in the result. You should not bring up new terms or information in your discussion

26. You should also focus on connecting the findings of the study to the study objectives. The objectives seem lost somewhere in the discussion

27. You should provide explanations and reasons behind each finding from the study

28. “These findings are similar to the studies conducted in Iran, USA, India, and Africa (80 83), where higher resistance rates were reported, highlighting the potential regional differences in antibiotic use and resistance patterns.” Please, provide more information on these studies.

29. You may also want to add some recommendations to the discussion

Conclusion

30. “Last resort” is this referring to the reserved group mentioned in the discussion section? If yes, ensure consistency in the terms used across the manuscript.

Thanks

Reviewer #4: Thank you for the opportunity to review this manuscript, which addresses an important public health issue in Tanzania antimicrobial resistance (AMR). The authors conducted a systematic review and meta-analysis assessing the prevalence and patterns of AMR across bacterial pathogens. The topic is timely, policy-relevant, and consistent with the WHO Global AMR Surveillance System (GLASS) priorities. Overall, the manuscript has scientific value, but several areas require strengthening to meet publication standards. Below is a detailed critique structured around the scientific components of the manuscript.

1. Introduction

Strengths

Gives a clear general background on antimicrobial resistance and global importance.

Cites authoritative sources including WHO, NFID, and global AMR estimates.

Identifies ESKAPE pathogens as critical threats and connects this to clinical relevance.

Areas for Improvement

a. Literature Review Depth

The introduction describes global and Tanzanian context but does not sufficiently review past systematic reviews or local prevalence meta-analyses. For example:

EMR surveillance reports (Camara 2023, Silago 2024) are mentioned later but not integrated into the introduction.

The review does not discuss earlier regional AMR systematic reviews beyond one sentence.

A stronger introduction should synthesize:

Regional AMR trends (East Africa 2005–2020)

Tanzania’s National AMR Action Plan (2017–2022)

Gaps in surveillance and empirical treatment guidelines (STG/NEMLIT)

b. Clear identification of the knowledge gap

The authors state that “comprehensive data remain sparse” but should articulate:

What has not been synthesized previously

Why a national pooled prevalence is needed now

How this research will directly inform Tanzanian AMR policy

c. Objective Placement and Clarity

The last paragraph provides an objective (“to present a thorough overview…”) but this should be formatted as a clear study aim:

“This systematic review and meta-analysis aimed to estimate pooled prevalence of antibiotic resistance among clinically relevant bacterial pathogens in Tanzania from 2014–2024 and evaluate resistance patterns across antibiotic classes using subgroup analyses.”

2. Methods

Strengths

Protocol registered on PROSPERO (major strength).

PRISMA guidelines followed appropriately.

Search terms and Boolean operators provided.

Clear inclusion and exclusion criteria.

Data extraction and quality assessment tools described.

Meta-analysis approach with random-effects model is appropriate given expected heterogeneity.

Areas for Improvement

a. Search Strategy

Google Scholar yielded 18,000 results. The authors should specify:

how many pages were screened

which filters were used

what strategies were used to avoid missing key grey-literature but also avoid irrelevant hits.

b. Duplicates

Only 416 duplicates removed out of 18,265 – this appears unusually low. Most systematic reviews typically remove 10–30% duplicates. Authors should clarify if:

deduplication was automated,

inclusion was restricted by year,

Google Scholar citations were sorted.

c. Data Extraction

A brief description is provided, but reproducibility requires more detail:

Was a data extraction form piloted?

How were discrepancies resolved between the two independent reviewers?

Was interrater agreement (κ statistic) assessed?

d. Risk of Bias Assessment

The manuscript states that JBI cross-sectional appraisal tools were used.

However:

The results of the appraisal should be summarized in the Results section.

A table (S1) is referenced but not fully interpreted or summarized in the text.

e. Meta-analysis Specifics

To ensure reproducibility, the authors must specify:

Thresholds for heterogeneity (I² cutoff)

Handling of zero-event studies

Whether proportions were transformed (logit, Freeman–Tukey double arcsine)

This is essential because meta-analysis of proportions is highly sensitive to transformation choice.

f. Ethics

Systematic reviews do not require IRB approval.

However, the authors should state this explicitly:

“As this study used published data only, institutional review board approval was not required.”

3. Results

Strengths

PRISMA flow diagram is clear and appropriate.

Characteristics of included studies are well organized.

Tables and figures illustrate study design, sample sources, and regional distribution.

Forest plots are visually clear and appropriately formatted.

Areas for Improvement

a. Chronological Alignment with Objectives

Results should be structured into clear subsections:

Description of included studies

Resistance patterns by antibiotic class

Resistance patterns by pathogen

Heterogeneity assessment

Currently, there is overlap between description and interpretation of findings.

b. Over-reporting raw proportions

Many values are repeated excessively in the narrative. For a meta-analysis, the text should highlight:

Most resistant pathogens

Most resistant antibiotic classes

Most consistent patterns across studies

Any observable trend over time

c. Figures and Tables

Some figure numbers overlap (Fig 1 appears referenced twice).

Figures S1–S14 are referenced but not described in adequate detail.

d. Missing Footnotes and Abbreviation Definitions

Table 1 uses acronyms (EMD, IPD, SSI, PROM) without footnotes. Standard reporting requires defining all abbreviations.

4. Discussion

Strengths

Adequately interprets overall findings.

Connects resistance patterns to clinical relevance and treatment guidelines.

Compares findings with regional and global literature.

Identifies implications for empirical prescribing guidelines.

Areas for Improvement

a. Too Much Repetition of Results

Several paragraphs simply restate the same numeric values from the Results section.

The Discussion should instead emphasize:

Why resistance is high

Mechanisms (antibiotic misuse, OTC availability, weak stewardship, hygiene gaps)

Regional differences

Policy impact

b. Missing Interpretation of Heterogeneity

High heterogeneity (>75%) was identified in most analyses but not discussed.

The authors must explain:

Possible causes: variation in study designs, regions, testing methods

Implications for pooled estimates

c. Lack of Forward-Facing Recommendations

Given Tanzania has a National AMR Action Plan, the authors should recommend:

Strengthening surveillance systems

Restricting OTC antibiotic access

Enhancing stewardship in hospitals

Improving laboratory diagnostics

Updating STG/NEMLIT guidelines

d. Missing Consideration of Publication Bias

Even though publication bias tests for proportion meta-analysis are limited, authors should state:

Why funnel plot asymmetry was not tested

Limitations related to publication bias

5. Limitations

This section is present but needs expansion to include:

High heterogeneity across studies

Overrepresentation of urban hospital data

Underrepresentation of rural facilities

Variation in laboratory testing standards

Possible publication bias

No time-trend analysis due to inconsistent reporting years

6. Conclusion

Strengths

Summarizes major findings.

Emphasizes importance of stewardship and surveillance.

Areas for Improvement

Should more clearly reflect the primary objective: pooled prevalence.

Should include concrete policy recommendations.

Should avoid adding new information not discussed earlier.

Should follow PRISMA standard: one paragraph summarizing findings, implications, and future needs.

SUMMARY FOR AUTHOR

To improve the manuscript for resubmission, the authors should:

Expand the introduction with stronger literature review and clearer objectives.

Provide a more reproducible and transparent search strategy.

Clarify duplicate removal and data extraction steps.

Explicitly describe meta-analysis transformations and heterogeneity interpretation.

Structure the results more clearly and reduce redundancy.

Strengthen the discussion with causal explanations, regional differences, policy implications, and stewardship recommendations.

Add a complete limitations section.

Improve clarity and grammar throughout.

Provide complete data availability in supplemental files.

Ensure all figures, abbreviations, and footnotes meet reporting standards.

Ensure all figures are accessible or in the paper.

.

Reviewer #1: **Yes:**Olga PerovicOlga PerovicOlga PerovicOlga Perovic

Reviewer #2: **Yes:**Uchechukwu Chiemelu, MHA, BSc., MLTUchechukwu Chiemelu, MHA, BSc., MLTUchechukwu Chiemelu, MHA, BSc., MLTUchechukwu Chiemelu, MHA, BSc., MLT

Reviewer #3: **Yes:**Mojirade M AThedetobaMojirade M AThedetobaMojirade M AThedetobaMojirade M AThedetoba

Reviewer #4: No

---

## [Author Response · Author response to Decision Letter 1]

16 Feb 2026

Our responses to all specific reviewer and editor comments have been provided as attachments for your review.

Thank you

---

## [Decision Letter · Decision Letter 1]

10 Mar 2026

Dear Dr. Kafaiya,

Thank you for submitting your manuscript to PLOS ONE. After careful consideration, we feel that it has merit but does not fully meet PLOS ONE’s publication criteria as it currently stands. Therefore, we invite you to submit a revised version of the manuscript that addresses the points raised during the review process.

We look forward to receiving your revised manuscript.

Kind regards,

Mabel Kamweli Aworh, DVM, MPH, PhD. FCVSN

Academic Editor

PLOS One

**Journal Requirements:**

**Additional Editor Comments:**

In addition to addressing the comments from Reviewer 2, please attend to the following minor issues:

Since **AMR** was defined in line 57, please use the abbreviation was defined in line 57, please use the abbreviation was defined in line 57, please use the abbreviation was defined in line 57, please use the abbreviation **AMR** in place of in place of in place of in place of *antimicrobial resistance* in subsequent occurrences (e.g., lines 79, 177, etc.). Please ensure that the use of this abbreviation is applied consistently throughout the manuscript.in subsequent occurrences (e.g., lines 79, 177, etc.). Please ensure that the use of this abbreviation is applied consistently throughout the manuscript.in subsequent occurrences (e.g., lines 79, 177, etc.). Please ensure that the use of this abbreviation is applied consistently throughout the manuscript.in subsequent occurrences (e.g., lines 79, 177, etc.). Please ensure that the use of this abbreviation is applied consistently throughout the manuscript.Line 184, add "." at the end of the sentence.Please remove the subheading **“Limitations.”** The study limitations should instead be incorporated into the final paragraph of the The study limitations should instead be incorporated into the final paragraph of the The study limitations should instead be incorporated into the final paragraph of the The study limitations should instead be incorporated into the final paragraph of the **Discussion** section.section.section.section.Please replace the subheading **“Conclusion, Recommendations, and Policy Implications”** with with with with **“Conclusions.”** Any recommendations and policy implications should be integrated into the final paragraph of the Any recommendations and policy implications should be integrated into the final paragraph of the Any recommendations and policy implications should be integrated into the final paragraph of the Any recommendations and policy implications should be integrated into the final paragraph of the **Conclusions** section.section.section.section.In line 361, please delete the abbreviation "(UTIs)" which was not used anywhere else in the manuscript and not listed under the list of abbreviations.

Reviewers' comments:

Reviewer's Responses to Questions

**Comments to the Author**

Reviewer #1: All comments have been addressed

Reviewer #2: (No Response)

Reviewer #3: All comments have been addressed

Reviewer #4: All comments have been addressed

2. Is the manuscript technically sound, and do the data support the conclusions?

Reviewer #1: Yes

Reviewer #2: Yes

Reviewer #3: Yes

Reviewer #4: Yes

3. Has the statistical analysis been performed appropriately and rigorously?

Reviewer #1: Yes

Reviewer #2: Yes

Reviewer #3: Yes

Reviewer #4: Yes

4. Have the authors made all data underlying the findings in their manuscript fully available?

Reviewer #1: Yes

Reviewer #2: Yes

Reviewer #3: Yes

Reviewer #4: No

5. Is the manuscript presented in an intelligible fashion and written in standard English?

Reviewer #1: Yes

Reviewer #2: Yes

Reviewer #3: Yes

Reviewer #4: Yes

Reviewer #1: I take this opportunity to thank authors for addressing reviewers comments appropriately. It is clear that the current review highlights the critical rates of antimicrobial resistance among ESKAPE-E pathogens in Tanzania. I have no further comments.

Reviewer #2: Reviewer Recommendation and Comments:

Manuscript Number PONE-D-25-41665R1

The integration of a quality assessment tool would be best discussed after data synthesis, analysis and reporting section.

Line 165-166 “Discrepancies between reviewers were resolved through discussion, and when consensus was not reached, a third reviewer adjudicated.” You can also state that the methodological quality of the included studies was independently assessed by two reviewers using the JBI Critical Appraisal Checklist for Prevalence Studies, and any discrepancies were resolved through consensus.

Limitations: You mention heterogeneity is "likely due to differences in... antimicrobial susceptibility testing (AST) practices". You could specify if this refers to different guidelines, such as CLSI vs. EUCAST, as these are common sources of variation in AMR studies. Additionally, the phrase "formal assessment of publication bias was not performed due to methodological limitations" could be more specific. You might mention that standard tests (like Egger's test) are often underpowered or inappropriate for proportions/prevalence meta-analyses.

Conclusion, recommendations, and policy implications: In the first sentence, there is a missing comma between "bloodstream infections" and "pneumonia”. Additionally, there is a typographical error in lines 367 and 368.

Reviewer #3: Thank you for revising the manuscript. It has greatly improved, and all previous comments have been addressed.

The only minor issue I noticed was with the abbreviations. Since there is a section defining abbreviations, I suggest removing the definitions from the other sections to avoid repetition.

Reviewer #4: The authors should be commended for their thorough and constructive engagement with reviewer feedback. The revised manuscript demonstrates substantial improvement across all previously identified areas of concern.

The statistical analysis is appropriate, transparent, and sufficiently rigorous for the study design.

Key improvements since the previous round include:

Explicit definition of dependent and independent variables and reference categories

Clear justification for batch-level analysis and the inability to perform multilevel modelling

Appropriate use of univariate and multivariable logistic regression

Reporting of model diagnostics (Hosmer–Lemeshow test and AUC)

Explicit discussion of collinearity, correlated predictors, and unequal seasonal sample sizes

The revised interpretation of seasonal effects particularly the attenuation and direction change after adjustment is now statistically sound and carefully explained.

In particular:

The Methods section is now comprehensive and reproducible, with clear variable definitions, data cleaning procedures, and modelling decisions.

Statistical inconsistencies identified in the previous round have been corrected, and interpretations are now aligned with the final model specification.

The addition of a choropleth map appropriately complements the tabular spatial analysis and supports descriptive claims without overstating inference.

The Discussion and Conclusion have been carefully revised to avoid causal language and speculative claims, with limitations clearly acknowledged and integrated.

Overall, the manuscript now provides a robust, transparent, and policy-relevant epidemiological description of bovine brucellosis patterns in Mpumalanga Province and represents a valuable contribution to the literature on routine surveillance data use in endemic settings.

I have no further substantive comments and support publication.

.

Reviewer #1: **Yes:**Olga PerovicOlga PerovicOlga PerovicOlga Perovic

Reviewer #2: **Yes:**Uchechukwu ChiemeluUchechukwu ChiemeluUchechukwu ChiemeluUchechukwu Chiemelu

Reviewer #3: **Yes:**Mojirade AdetobaMojirade AdetobaMojirade AdetobaMojirade Adetoba

Reviewer #4: No

---

## [Author Response · Author response to Decision Letter 2]

18 Mar 2026

We have reviewed carefully all comments provided by the academic editor and reviewers. Point-by-point responses to each comment have been prepared and are included in the document titled “Response to Reviewers,” which has been submitted as a separate file. In this document, we clearly indicate how each comment has been addressed and specify the corresponding revisions made in the manuscript.

---

## [Editor Report · Decision Letter 2]

19 Mar 2026

Prevalence of Antimicrobial Resistance in Tanzania: A Systematic Review and Meta-Analysis

PONE-D-25-41665R2

Dear Dr. Kafaiya,

We’re pleased to inform you that your manuscript has been judged scientifically suitable for publication and will be formally accepted for publication once it meets all outstanding technical requirements.

Kind regards,

Mabel Kamweli Aworh, DVM, MPH, PhD. FCVSN

Academic Editor

PLOS One
---

## [Editor Report · Acceptance letter]

PONE-D-25-41665R2

PLOS One

Dear Dr. Kafaiya,

I'm pleased to inform you that your manuscript has been deemed suitable for publication in PLOS One. Congratulations! Your manuscript is now being handed over to our production team.

Kind regards,

on behalf of

Dr. Mabel Kamweli Aworh

Academic Editor

PLOS One